# BENCHMARKING IN-CONTEXT EXPERIENTIAL LEARNING THROUGH REPEATED PRODUCT RECOMMENDATIONS

## ABSTRACT

To reliably navigate ever-shifting real-world environments, agents must grapple with incomplete knowledge and adapt their behavior through *experience*. However, current evaluations largely focus on tasks that leave no ambiguity, and do not measure agents' ability to adaptively learn and improve as they accrue experience. We exemplify the need for in-context experiential learning in a product recommendation context, where agents must navigate shifting customer preferences and product landscapes through natural language dialogue. We curate **BIEL**: a benchmark that combines i) rich real-world products from Amazon, ii) a diverse collection of user personas to represent heterogeneous yet latent preferences, and iii) a LLM user simulator powered by the persona to create realistic and interactive trajectories. We observe that current frontier models struggle to meaningfully improve across episodes, underscoring the need for agentic systems with strong in-context experiential learning capabilities.

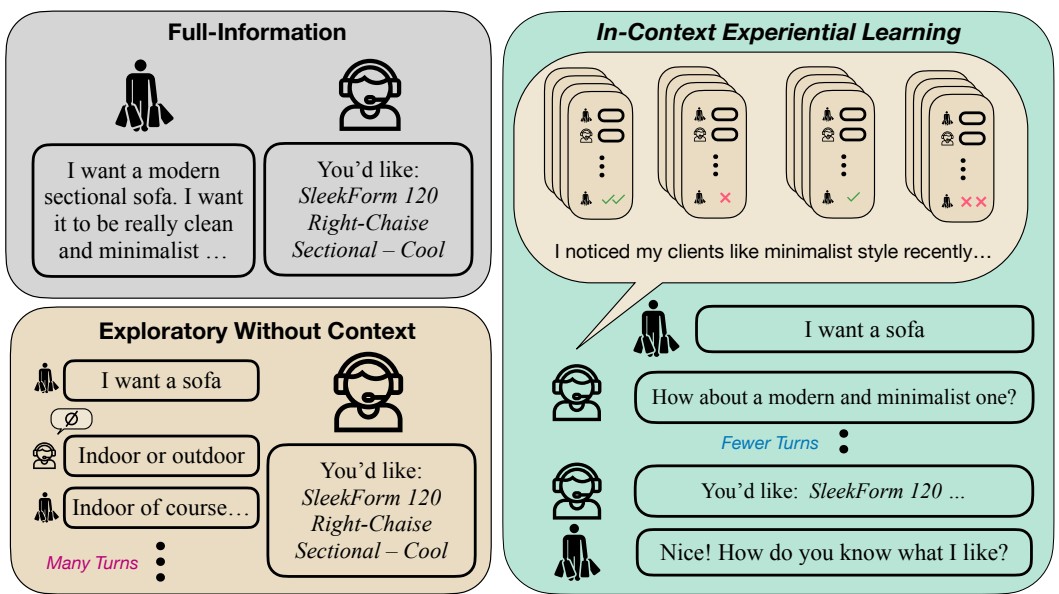

Figure 1: **Top Left**. Typical agentic benchmarks (e.g. Yao et al. (2024)) focus on settings where all information is provided initially, and the model is tasked with producing the correct answer in a zero-shot fashion. **Bottom Left**. Recent benchmarks (e.g. Li et al. (2024)) for LLM agents increasingly focus on multi-turn settings. **Right**. Our In-Context Experiential Learning setting.

---

**Examples of Recommendation Dialogues for In-Context Experiential Learning**

**Ep 1 (Hair Styling Gels):**

**AGT** : What is your hair type?

**CUST 1**: Curly.

**AGT** : What level of hold?

**CUST 1**: Medium.

**AGT** : What is your budget range?

**CUST 1**: Around $10 to $15.

**AGT** : Do you prefer a gel that provides moisture?

**CUST 1**: I prefer a gel that [...]

**Selected Product:**
ID: 34444
*Frizz Free Styling Smoother*

**CUST 1 Feedback:**
I appreciate the suggestion, but I was actually looking for something more focused on care, like a conditioning balm or oil to keep it healthy and styled.

**Ep 2 (Packaged Rice Dishes):**

**AGT** : What's your budget range for packaged rice dishes?

**CUST 1**: I usually aim to keep it under $10 per package.

**AGT** : Do you have any dietary restrictions?

**CUST 1**: I prefer a plant-based diet.

**AGT** : Do you prefer specific rice varieties?

**CUST 1**: I prefer wild rice.

**Selected Product:**
ID: 72329
*Dr. McDougall's Vegan Pilaf*

**CUST 1 Feedback:**
Oh, this looks great! I really appreciate the suggestion, but I usually prefer something with more whole grains.

**Ep 3 (Almonds):**

**AGT** : Do you prefer raw or roasted almonds, or flavored varieties?

**CUST 2**: Raw almonds.

**AGT** : What are your preferences for this product category?

**CUST 2**: I appreciate products that are sustainable, environmentally friendly, and ethically sourced.

**Selected Product:**
ID: 72479
*Roasted Sugared Almonds*

**CUST 2 Feedback:**
Oh dear, I was really hoping for something more in line with sustainable and locally sourced goods. This recommendation seems a bit off.

Figure 2: **Benchmark for In-context Experiential Learning (BIEL):** An exemplar recommendation dialogues for in-context experiential learning across 2 customer personas and 3 choice sets.

# 1 INTRODUCTION

The ability to learn and improve from experience is a hallmark of intelligence. Real-world environments involve uncertainty arising from unobserved information, and intelligent agents must deliberately act to minimize mistakes and quickly learn from experience. However, the prevailing pre- and post-training paradigms primarily focus on knowledge distillation (Brown et al., 2020; Christiano et al., 2017; Stiennon et al., 2020; Ouyang et al., 2022; Guo et al., 2025); while incredibly effective at tasks with little uncertainty (e.g., instruction following, math exams), resulting models often lack the ability to grapple with uncertainty, let alone to improve through repeated interactions with the environment (Liu et al., 2024; Zhou et al., 2024).

Following Silver & Sutton (2025), we refer the ability to adapt and improve from heterogeneous past interactions as in-context *experiential learning*. Without this ability, agents are confined to solving only familiar, fully-observable problems, leaving them brittle and ill-equipped to handle real-world tasks shaped by ever-changing environments and new uncertainties. Resilient and reliable agentic systems must be capable of long-horizon planning involving actively gathering costly yet informative feedback to reduce future uncertainty, reassessing uncertainty based on the feedback ("posterior updates"), and refining strategies over time.

We propose and construct a benchmark that measures the agent's ability to reason through uncertainty, and make discoveries over time by leveraging past interactions / episodes. We exemplify in-context experiential learning capabilities using recommendation tasks (Figure 1), which offer a naturally dynamic environment characterized by a constant stream of new customers and products. An effective agent must actively discover users' latent preferences through exploratory questions and iteratively refine recommendation based on ambiguous, text-based feedback. Notably, departing from the common formulation of partially observable Markov decision process (POMDP), we consider the "rewards" primarily encoded in *free-form natural language responses* (Yuksekgonul et al., 2025). The difference highlights a key requisite for a capable LLM agent: the ability to interpret and learn from natural language feedback.

| Aspect | MediQ | Streambench | LMRL Gym | Science World | BIEL (ours) |
|---|---|---|---|---|---|
| Multi-turn | ✓ | ✗ | ✓ | ✓ | ✓ |
| Exploratory | ✓ | ✗ | ✗ | ✓ | ✓ |
| Scalability | ✗ | ✗ | ✗ | ✗ | ✓ |
| Experiential | ✗ | ✓ | ✗ | ✓ | ✓ |

Table 1: Key axes evaluated across benchmarks. ✓ = satisfies, ✗ = does not, ✗ = mixed.

Recommendation tasks provide a fertile testbed for in-context experiential learning capabilities since each new customer and product introduces fresh *uncertainties*: the customers' preferences are initially unknown, and new sets of products form unseen landscapes of available choices. An ideal recommender agent must actively plan its interactions, strategically select questions to elicit responses over multiple turns, and eventually provide a final recommendation. Consider two representative3 scenarios. In the first scenario ("personalization"), the agent must discover and attend to a particular user's preference as it recommends different products and receives feedback over time. In the second scenario ("choice set"), the agent repeatedly sells a fixed choice set to a rotating pool of new customers and the focus shifts to learning how the products compare to each other across a diverse customer distribution.

In both of these scenarios, we envision agents that can actively discover user preferences through experience by leveraging multiple episodes. See Figure 2 for an example of our setting. Compared to works on pluralistic alignment or uncertainty quantification that focus on a single interaction/episode (Castricato et al., 2025; Zollo et al., 2025; Li et al., 2024), we primarily focus on the ability to learn across multiple interactions (*experiential learning*). Even when restricted to the personalization setting, we are interested in an agent that continuously interacts with the same customer, where the task is to gradually uncover and tailor the agent's recommendation to the customer's preferences.

We curate a large dataset for language-based recommendations by first pulling Amazon products from Hou et al. (2024) and categorizing them into a predefined list of categories (ASINSpotlight, 2023). This categorization enables us to form sets of interchangeable products that a customer might consider during a shopping session. Next, we draw on persona descriptions from Li et al. (2025) and use an LLM to simulate user preferences over these products. Simultaneously, the user-simulating LLM form the backbone of an interactive question-answering module, enabling realistic and dynamic interactions between agents and simulated users.

Beyond serving as a benchmark for evaluating experiential learning capabilities, our dataset offers a versatile and realistic framework for studying recommendation systems. By combining real-world product data, diverse user personas, and interactive dynamics powered by LLM simulator, it enables researchers to explore a wide range of questions including user modeling, preference elicitation, and cold-start recommendation, to name a few. We believe our dataset can be of separate and significant interest to the broader recommendation systems community.

Our main contributions are as follows:

- We formulate in-context experiential learning as a centerpiece to intelligence. Agents must be able to implicitly reason through uncertainty and refine their strategies by leveraging past experience.

- Going beyond sequential tool-use capabilities, we focus on multi-episodic settings where the agent necessarily have to mistakes initially due to ambiguity. Key differences from prior benchmarks are summarized in Table 1.

- We develop a Benchmark for In-context Experiential Learning (BIEL), a dataset of diverse and scalable product categories, including 71K products and 2K choice sets. Coupled with 1M scalably generated personas from Li et al. (2025), we support up to 2B multi-turn environments on which one can build a wealth of experiential learning settings.

- We observe even state-of-the-art models generally fail to exhibit meaningful learning across episodes, highlighting their inability to navigate ever-shifting real-world environments.

## 2 RELATED WORKS

**RL for Language Model Training.** The primary application of reinforcement learning (RL) in large language model (LLM) training has been Reinforcement Learning with Human Feedback (RLHF) (Christiano et al., 2017; Stiennon et al., 2020; Ziegler et al., 2020). RLHF has proven highly effective for aligning models with complex, difficult-to-quantify objectives that lack well-defined, differentiable reward functions: for example, RLHF has enabled models to improve on dimensions such as translation quality (Ramos et al., 2024; Kreutzer et al., 2018), helpfulness (Ouyang et al., 2022; OpenAI et al., 2024), and factual accuracy (Bai et al., 2022; Glaese et al., 2022; Touvron et al., 2023; Sun et al., 2024), among other desiderata. However, these common settings are fundamentally limited: they correspond to environments that consist of only a single-step interaction, where the model takes one action (i.e. generating a response), and immediately receives a scalar reward. While this paradigm has been remarkably successful for training high-performing zero-shot models, it fails to capture the interactive, dynamic nature of real-world decision-making, where actions and feedback over multiple turns or episodes can aid the models' decisions. In contrast, our work focuses on multi-turn and multi-episode settings, and highlights the insufficiency of current state-of-the-art models to adapt and improve through interactions.

**Multi-turn and Multi-episode RL.** Recent years have witnessed a surge of interest in agentic models powered by LLMs (Jimenez et al., 2024; Yao et al., 2024; Karten et al., 2025). These agents are expected to autonomously plan, act and adapt through iterative interactions with their environment, requiring models to engage in *multi-turn interactions* within each episode, and to learn from experiences accumulated *across many episodes*. As a result, prior works have explored multi-turn settings in text-based games (Abdulhai et al., 2025; Tajwar et al., 2025), medical question-answering (Li et al., 2024), and numerous other tasks (Liu et al., 2024). A few other works have explored multi-episode settings (Wu et al., 2024; Zheng et al., 2025), but they primarily focus on environments devoid of uncertainties. Our experiential learning setting is closest to ScienceWorld (Wang et al., 2022), in which learning casual abstractions across episodes (Majumder et al., 2023) is the key to solving the tasks therein. In contrast, our setup focuses on assessing the ability of the models to *reason through uncertainties* of the latent preference of the customers in context.

**Recommendation System and LLM Personalization.** There has been a long line of work on recommendation systems (Resnick et al., 1994; Koren et al., 2009). In the era of foundation models, LLMs have demonstrated remarkable zero-shot performance in recommendation tasks (Geng et al., 2022; He et al., 2023; Lyu et al., 2024). However, the classical yet crucial setting of sequential recommendation (Hidasi et al., 2016; Tan et al., 2016) remains underexplored in this era, and the even more realistic problem of modeling interactions with customers has received little attention. Our dataset addresses this gap by enabling the study of these settings through an interactive user simulator powered by LLMs. Separate but relatedly, the rise of LLMs has sparked growing interest in personalized LLMs (Castricato et al., 2025; Jang et al., 2024; Zollo et al., 2025). Compared to these post-hoc heuristic approaches, we propose a more principled way to instill agents with the ability to learn to personalize. Our dataset is designed to directly evaluate and help advance this capability.

## 3 RECOMMENDATION SYSTEMS AS A FERTILE TESTBED FOR EXPERIENTIAL LEARNING

Recommendation systems provide a rich foundation for testing the experiential learning capabilities of agentic systems. Consider a common shopping scenario: a *customer* ($c$) enters a store in seek of a product to satisfy a specific need. Many products fulfill the same functional purpose, but they differ in style, aesthetics, or price, leading to a preference unique to the customer. We refer to this group of functionally equivalent products as a *choice set* ($S$). The recommender agent must discover customer's preferences over $S$ by engaging in *multiple turns* of queries: asking targeted questions, receiving, and interpreting the customer's response. This entire interaction, from initial inquiry to final recommendation, constitutes an *episode*.

Following each recommendation, the agent typically receives *feedback*, such as a purchase decision or free-form text opinions about the suggested product. This feedback, combined with the intermediate responses, encodes rich information about the *latent factors* ($\theta$) underlying a given episode.

This latent factor may include the customer's preferences, the dynamics of the question-answering process, how the customer perceives the structure of the choice set, and so forth. Compared to a typical sequential decision-making setup (e.g., POMDPs), the "reward" based on which the agent should optimize its strategy is encoded primarily in text as part of the observations (Yuksekgonul et al., 2025). A capable agent must interpret these text-based signals to sharpen its belief of the latent $\theta$ to improve its recommendations in subsequent episodes.

A performant recommendation system must learn across users and products and improve based on experience. Thus, we are not merely interested in the recommender agent's performance within a single episode, but rather their ability to improve as its experience accrues. For instance, consider a personalized recommendation setting where the same customer interacts with the agent across multiple episodes. In this case, the agent should focus on uncovering the customer's underlying preferences to improve future recommendations. Conversely, imagine a scenario where the agent repeatedly sells a fixed choice set of products to a stream of new customers. Here, the goal shifts to identifying how these products compare relative to one another across the diverse distribution of customers. In the most difficult setting, both customers and choice sets can evolve over time.

Formally, the agent encounters a *stream of episodes* over its lifetime, where each $e$-th episode is characterized by a tuple $(c_e, \boldsymbol{S}_e)$ that induces a corresponding latent factor $\theta_e$. Each episode, indexed by $e = 1, ..., E$, represents a single shopping session defined by a tuple $(c_e, \boldsymbol{S}_e)$, representing a customer and a choice set of products. In an episode (shopping session), customer $c_e$ is interested in buying one product from $\boldsymbol{S}_e$ according to their preference. Each episode contains a sequence of turns where each turn consists of the recommender's question or recommendation $a_{e,t}$, and the customer's response $o_{e,t}$. If the action $a_{e,t}$ is a question, then the response $o_{e,t}$ is the customer's answer; if $a_{e,t}$ is a recommendation, then $o_{e,t}$ is a feedback provided by the customer, encoding the customer's preference over the recommended product. Naturally, the $e$-th episode concludes after a recommendation is made, and the task moves on to the next episode. We denote this final feedback as $f_e$ for convenience.

We denote the the sequence of latent factors by $\boldsymbol{\Theta} = \{\theta_e | e = 1, 2, ...\}$; some components of these latent factors may remain stable over time, while others may shift. An intelligent agent must learn to adapt to the dynamics of $\boldsymbol{\Theta}$ in an online fashion. At each turn, the response $o_{e,t} = g(\theta_e, a_{e,t})$ is a function of both the action and the latent factor of the episode. At episode $e$ and turn $t$, the entire history available to the recommender agent is

$$\mathcal{H}_{e,t} = \underbrace{\left\{ (a_{e',t'}, o_{e',t'}) | \, \forall e' \in [e-1], \forall t' \in [T_{e'}] \right\}}_{\text{Past Episodes}} \cup \underbrace{\left\{ (a_{e,t'}, o_{e,t'}) | \, \forall t' \in [t-1] \right\}}_{\text{Current Episode}},$$

with which the agent $\pi(\cdot)$ takes an action $a_{e,t} = \pi(\mathcal{H}_{e,t})$.

To measure the quality of the recommendations, we score the products for each unique pair of $(\boldsymbol{S}, c)$ that defines an episode. Given one such pair, for each product $p_a \in \boldsymbol{S}$, we require a score $y_a = h(p_a, c)$. The main metric of merit for the recommendation is the *regret*: $y^* - y_r$, where $y^*$ is a highest score, $y^* = \max \left( \{ y_a | \, y_a = h(p_a, c, \boldsymbol{S}) \, \forall p_a \in \boldsymbol{S} \} \right)$, and $y_r$ is the score of the recommended product.

## 4 BENCHMARK FOR IN-CONTEXT EXPERIENTIAL LEARNING (BIEL)

To instantiate the above experiential learning framework, our benchmark comprises of two main modules: (1) a set of $\boldsymbol{C} = \{c_i\}$ of hypothetical customers that can respond *interactively* to the recommender, and (2) a set $\mathcal{S} = \{\boldsymbol{S}_j\}$ of choice sets. For both modules, we want the underlying sets to be scalable and diverse to support nuanced scoring and a wide variety of $\boldsymbol{\Theta}$ dynamics. We address this challenge by designing novel filtering schemes that allow us to construct a large-scale language-based product recommendation dataset, consisting of 71K products and 2K choice sets. (Since some categories share products, the dataset contains in total 100K product-category pairs.)

**Persona-Based Customer Simulation.** To power the customer module, we use the 1M persona specifications in Li et al. (2025) to provide diversity in $\boldsymbol{C}$. We then utilize LLMs to simulate the customer's response corresponding to a persona file to enable interactive question-answering. The persona therein are generated by LLMs, and hence are as scalable and diverse as what modern LLMs encode.

---

**An Example Persona and Their Preferences**

Meet Karen Thompson, a 59-year-old woman living in Minneapolis, Minnesota. She has short, curly brown hair and bright blue eyes, often accentuated by her warm and welcoming smile. [...]

**Simplified Preference Scores (Hair Styling Gels)**

| ID | Product | Score |
|----|---------|-------|
| 40255 | Tigi Bed Head Foxy Curls Contour Creme, 6.76 oz | 95.0 |
| 34764 | Shea Moisture Sacha Inchi Oil Curl Defining Smoothie, 12 oz | 85.0 |
| . . . | . . .    *more products*    . . . | . . . |
| 35903 | NKICAW IVILA Hair Straightening Cream (3PCS) | 15.0 |

---

Figure 3: An exemplar persona taken from Li et al. (2025) and their preferences over a category of products from Hou et al. (2024). Scoring is done by GPT-4o and Gemini-1.5-Pro. Consistent with the persona's curly hair, curl-enhancing products are rated highly, whereas straightening products receive low scores.

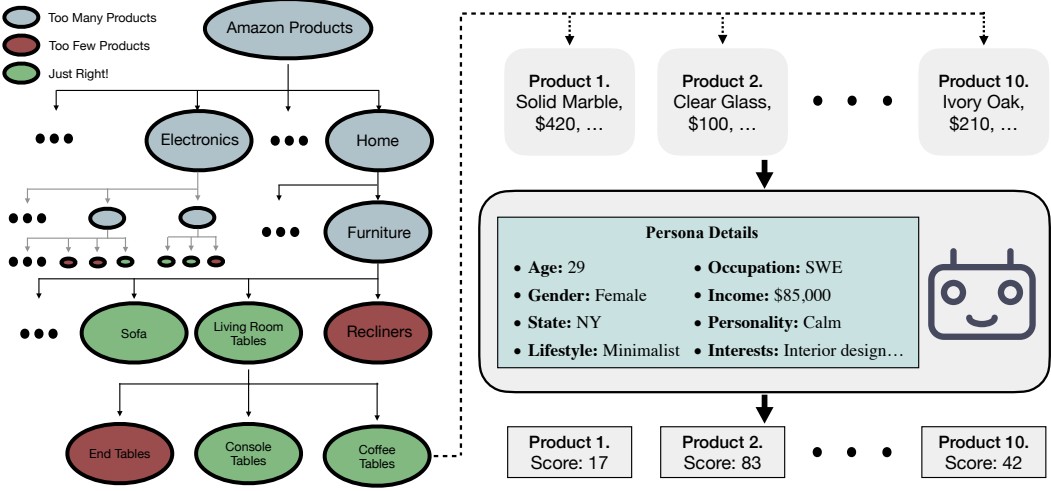

Figure 4: We use a predefined tree of categories (ASINSpotlight, 2023) and filtered the ones unsuitable as choice sets. We then score the products within each choice set with a persona-simulating LLM.

**Products and Choice Sets Curation.** To form the choice sets module, we need a pool of products, correctly categorized into reasonable choice sets. For the pool of products, we sampled 100K products from the Amazon Reviews dataset (Hou et al., 2024), which consists of 34 primary product categories. We applied filters to remove products missing essential fields such as product descriptions. We match each product to a predefined category from ASINSpotlight (2023). Through a rigorous matching process, we were able to assign 70% of the sampled products to the categories of ASINSpotlight (2023); see Appendix C for additional details.

Then, we extract $\mathcal{S}$ as a subset of predefined categories, consisting of categories that could reasonably be treated as choice sets. We first truncate the top two levels of this category hierarchy as these levels contained overly broad classifications ill-suited as choice sets (e.g., Home & Kitchen). We also filter categories that contain too few products to get nontrivial recommendation settings. See Figure 4 for an illustration of the process.

On scalability of available choice sets, one can easily imagine expanding the number of valid choice sets by sampling additional products from Hou et al. (2024) or other large-scale datasets. to increase the number of valid choice sets using our pipeline. Moreover, our framework is designed to be broadly compatible, allowing it to incorporate any choice set that contains a sufficient number of products with descriptions. More sophisticated or specialized data collection pipelines could be developed to gather richer and more diverse choice sets, further extending the scope of our benchmark.

**Preference Generation.** For preference generation ($y_a = h(p_a, c, \boldsymbol{S})$), the scoring function $h(\cdot)$ was an average of the scores generated using two models: GPT-4o and Gemini-1.5-Pro. Each model received the full description of the current persona encoding $c$, along with details of the products in the set $\boldsymbol{S}$ under evaluation. The models were instructed to assign a score (0–100) reflecting how much the persona would enjoy or appreciate each product within $\boldsymbol{S}$. The consistency of the scoring process is crucial to the integrity of the benchmark. To evaluate the consistency, we randomly selected five categories and measure the average variation in the scores across three seeds. We found that the averaged score variation is $4.1$, much smaller than the standard deviation in scores within each category ($25.4$), which suggests that the scoring process is consistent. For instance, in Figure 3, the persona is described as having *curly brown hair*. Consistent with this trait, products intended for curl enhancement are assigned high scores, while products designed for hair straightening receive lower scores.

**Customer Simulation and Feedback.** Users are simulated by a LLM (GPT-4o) prompted to act as the persona when shopping within a given product category. To prevent unrealistic scenarios (e.g., an individual with no musical interests shopping for pianos), we exclude categories in which the highest product score is below 60. Using this filtering procedure, an average of $3.75$ categories were skipped across the first 20 seeds.

**Performance measures.** At the end of each $e$-th episode, the simulated user provides a feedback $f_e$. We support three forms of feedback: (1) **Regret**, (2) **Stars**, and (3) **Free-form Text**. Regret feedback directly provides the regret of the recommendation. While informative, this metric is unrealistic in practice, as it is typically infeasible to obtain exact utility scores from real users. To approximate more practical feedback, we provide star ratings as $5 \times \frac{\text{score of chosen product}}{\text{score of best product}}$, rounded to the nearest integer, mirroring the star systems commonly found on e-commerce platforms. Finally, free-form text feedback involves prompting the customer LLM with the recommendation, its regret value, and the top three scored products, and instructing it to generate natural language feedback—expressed in the persona's voice on the extent to which the chosen product meets their preferences. See Figure 2 for examples. The generated feedback reflects the persona's curly hair type and consistently maintains the persona's character throughout the interaction.

**Robustness check.** To rule out the possibility that poor agent performance stems from an impossible task, we conducted a manual, human-curated questioning run as seen in Appendix A.5. The questions were deliberately *reasonable* in that they (i) targeted concrete, product-determining attributes, (ii) prompted the persona to reveal personality details, and (iii) were answerable without specialized knowledge. Under this regime, the agent identified the best scoring product (regret = 0) when it had otherwise consistently chosen a product with regret 37.5. This demonstrates that, given a well-planned query policy, the information required to reach the best item is accessible, and the environment is *solvable*.

## 5 EXPERIMENTS

We study three sequential recommendation settings that exemplify experiential learning: (1) same customer across episodes with different categories each episode, (2) same category across episodes with different customers each episode, and (3) different categories and customers each episode. Of course, the rich set of personas and categories available can support many more settings than the ones considered in this section.

To evaluate performance, aside from the aforementioned regret metrics, we monitor the number of questions asked. In our experiments, recommender agents are not explicitly prompted to opt for fewer questions. Therefore, they should ask as many questions as needed to resolve uncertainties for recommendation. We consider three baselines throughout. RANDOM recommends products uniformly at random, and POPULARITY selects the product with the highest original rating, breaking

ties by choosing the cheaper option. Finally, ORACLE is a Claude-Sonnet-4 with access to the full persona description and prompted to directly select a product from the category. Since Claude is not used during the initial scoring of products, this setup ensures a clean separation and serves as a lower bound on regret in cases where the recommender has full access to persona information.

## 5.1 COMPARING BASE MODELS

We first evaluate five models—GPT-4o, Gemini-2.5-Pro, Gemini-2.5-Flash, Claude-Opus-4, and Claude-Sonnet-4—over 10 episodes across 40 random seeds. Across these experiments, we use the Free-form Text feedback.

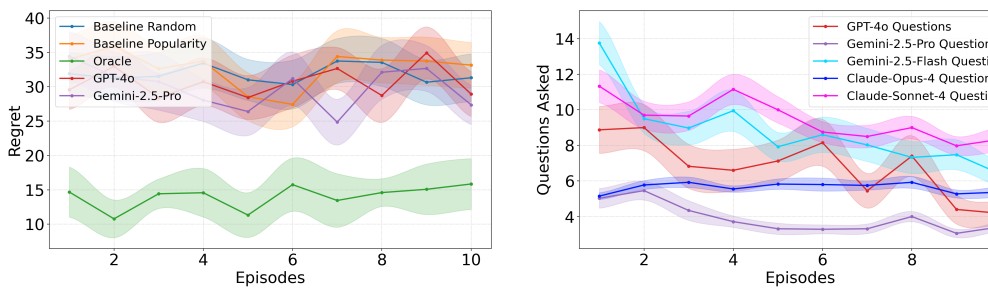

Figure 5: **LEFT**: The models are not learning from previous experiences. **RIGHT**: The models tend to ask fewer questions in later episodes, despite failing to learn from the interactions.

As shown in the left panel of Figure 5, none of the models achieve meaningful improvements over simple baselines, and all significantly underperform the oracle baseline. This highlights the difficulty of leveraging prior episodes for better recommendations for current SOTA models. Full ablations can be found in Appendix B.

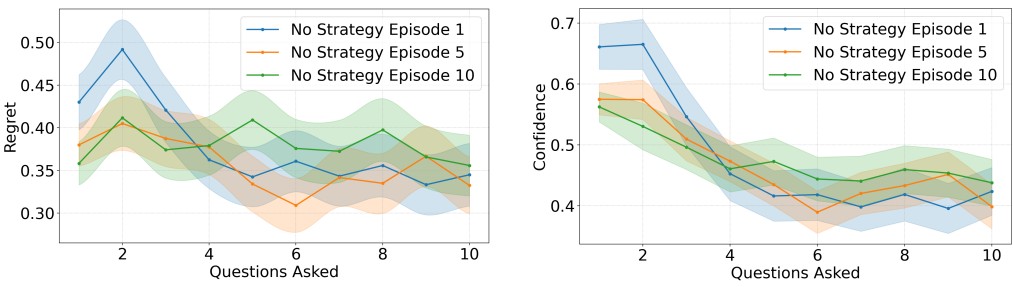

Figure 6: **LEFT**: GPT-4o performs no better across episodes nor over questions asked. **RIGHT**: GPT-4o grows less confident with more questions asked. Results are shown for confidence on regret within 10.

If agents perform poorly, we expect them to ask more questions to improve their recommendations. However, on the right of Figure 5, we observe that the number of questions asked generally declines across episodes. We highlight this as a major deficiency in the behavior patterns of the SOTA models.

To understand how models implicitly quantify its own uncertainty, we prompt the GPT-4o to output six kinds of confidences on: (1-2) whether the chosen item would fall into top-1 and top-5 favorite categories of the customer, and (3-6) whether the realized regret would fall into the expected regret within 5, 10, 20, and 30 points. See Figure 7. Across all settings, we found that the model is largely poorly calibrated. See Appendix B.5 for complete statistics.

An occasional issue with GPT-4o was an endless loop of repetitive questions. At times, the model would repeatedly ask the same queries despite having access to the full conversation history, continuing until the 20-question limit was reached. Increasing the temperature somewhat reduced the

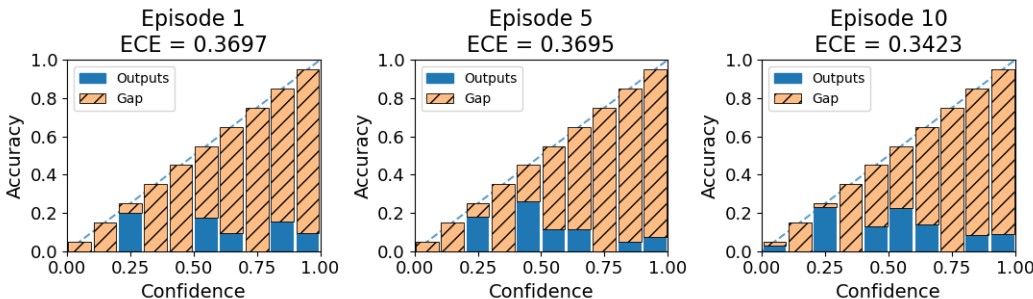

Figure 7: ECE plots for GPT-4o with No Strategy. The model is clearly poorly calibrated. Results are shown for confidence on regret within 10 across all turns.

frequency of this behavior, but did not eliminate it entirely. We observed similar conversational looping with other models as well; both Claude-Sonnet-4 and Gemini-2.5-Flash occasionally fell into these repetitive patterns.

## 5.2 PLANNING CAPABILITIES VIA PROMPTING

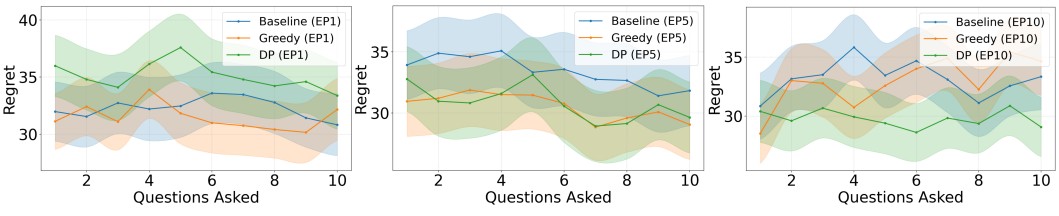

Figure 8: Performance of different prompting strategies across episodes (from left to right) with GPT-4o. Observe that only DP prompts seem to induce learning across episodes, whereas other prompts fail to improve.

We investigate whether explicit prompting strategies could encourage more deliberate planning in recommendations. We consider three variants were evaluated using GPT-4o over 60 random seeds: (a) simply prompts the agents to ask questions for $T$ turns (**No Strategy**), (b) explicitly instructs to act greedily (**Greedy**), and (c) instructs to act as a DP-optimal planner in a POMDP (**DP**). In all planning experiments, the agent was run for 10 episodes under standard conditions, with additional evaluations conducted in the 1st, 5th, and 10th episodes. In these episodes, the agent was additionally prompted to make a recommendation after each question, allowing us to measure regret at every turn. Figure 8 reports regret as a function of the number of questions asked. Greedy prompting occasionally yields some reductions in regret relative to the baseline, but does not improve across episodes. DP-style prompting exhibits greater improvements over episodes; by episode 10, it shows a reduction in regret compared to both the baseline and greedy prompting. This shows that prompting may encourage agents to learn, but the performance is nonetheless far worse than ORACLE.

## 6 CONCLUSION

We argue for a shift in focus toward measuring agents' ability to engage in adaptive, multi-episode interactions. To support this goal, we introduced a sequential recommendation dataset designed to evaluate in-context experiential learning capabilities. Beyond its use for benchmarking agents, our dataset may also be of independent interest to the broader recommender systems community.

Our experiments reveal that SOTA models do not learn from experience. They struggle to calibrate their uncertainties and, as a result, fail to proactively ask informative questions. By highlighting these limitations, we aim to motivate future research toward improving these critical abilities, which are essential for building reliable, real-world agents.

## 7 REPRODUCIBILITY STATEMENT

We provide the code used for all the experiments in this paper in `BIEL.zip` of https://github.com/17my15/BIEL/releases.

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

# A PROMPTS

## A.1 ORACLE AGENT

We present the prompt used to define the oracle baseline. The oracle agent is given the full persona description and the complete set of candidate products, and is instructed to return only the index of the single best product. This setting establishes a lower bound on regret under full-information conditions.

---

**Prompt for Oracle Agent Baseline**

**System Message:** You are an oracle recommendation agent with perfect knowledge of a customer's preferences.

**Customer Persona:**
`{self.persona_description}`

**Product Category:** `{category}`

**Available Products:**
`{self._format_products(products)}`

**Task:** Given the customer's complete persona description, choose the single best product that would most satisfy their preferences and needs. You have perfect knowledge of what this customer would want.

**Output format (MUST be exactly one line, no extra text):**
`RECOMMEND: <array_index_0_to_{num_products-1}>`

**Rules:**

- Choose the product that best matches the customer's persona.
- Consider all aspects of their preferences, lifestyle, and needs.
- Return the array index (0-based), not the product ID.
- No explanations, just the recommendation index.

---

Figure 9: Prompt used to instantiate the oracle baseline.

## A.2 PERSONA AGENTS

We introduce the prompts used for the persona agents. These agents are employed in three settings: (1) scoring personas, (2) generating dialogue, and (3) producing persona-based feedback.

---

**Prompts for Customer Simulation**

**(a) Preference Generation**
```
"persona_des" : "..."
"category"    : "..."
"products"    : "[...]"
"instructions": "You ARE the persona de-
               scribed.  Rate each product
               with a score from 0 to 100
               (integers only) based on how
               much YOU would like it. Re-
               turn a JSON object with key
               'results' as an array of objects:
               {id, score}.  Do not include
               any other keys or text."
```

**(b) Response Simulation**
```
"persona_des" : "..."
"question"    : "..."
"instructions": "You simulate a user with the
               given persona description. An-
               swer strictly as this persona
               would: – Only answer the
               question asked. – Do not re-
               state persona or add rationale.
               – If a choice is requested, give
               one choice only.  Return the
               answer as plain text."
```

---

Figure 10: Prompts for simulating customer behavior with persona agents: (a) generating product preference scores, (b) producing persona-consistent responses to agent queries.

---

**Prompt for User Feedback Response**

Reminder: You are a user with this persona:
`{self._persona_text}`

A recommendation agent just suggested a product to you.

**Context:**

- `{chosen_info}`
- `{conversation_context}`

`{tone_instruction}`

**Task:** Respond naturally as this persona would—like you're talking to a helpful salesperson or friend. Be conversational and specific about your preferences. Keep it to 1–2 sentences and sound like a real person, not a formal review. Make it a statement about your preferences, not a question. Never mention specific scores, regret values, or reveal which product would be better.

**Your response:**

---

Figure 11: Prompt for eliciting naturalistic persona feedback following a recommendation. The tone is adjusted based on the quality of the recommendation, estimated through regret.

## A.3 RECOMMENDER AGENT

We describe the prompts used to instantiate the recommender agent. At its core, the agent is queried at each turn with a baseline prompt, shown in Figure 12, which specifies the available context and requires the model to either ask one clarifying question or make a recommendation.

To evaluate whether agents can exploit past interactions, we additionally replace the raw conversation history with a summary of prior episodes, producing the variant shown in Figure 13.

To test whether explicit reasoning instructions improve performance, we append a chain-of-thought style enhancement to the baseline prompt, shown in Figure 14. These prompt designs define the controlled conditions under which we ablate model behavior, with results reported in Appendix B.

---

**Prompt for Recommender Agent Action**

**System Message:** You are a product recommendation agent. Your goal is to find the best product for this user, while asking the fewest number of questions before being confident in the best product for the user.

**Context:**
`{context}`
`{feedback_context}`

**Task:** First, analyze what you already know from the conversation. Then, either:

- Ask one short, consumer-friendly question to clarify user preferences, or
- If sufficiently confident, recommend one product by index.

**CRITICAL OUTPUT FORMAT (MUST FOLLOW EXACTLY):**

- To ask a question: `QUESTION: [your question here]`
- To recommend: `RECOMMEND: [number between 0 and {num_products-1}]`

**STRICT RULES:**

- Your response must start with either "`QUESTION:`" or "`RECOMMEND:`".
- Do NOT include any explanations, reasoning, or additional text.
- Do NOT use bullets, multiple lines, or formatting.

---

Figure 12: LLM prompt for the recommender agent to decide its next action (ask or recommend).

---

**Prompts for Episode Generation Summary**

**System Message:** You just completed Episode {episode_num} in the {category} category for Persona {persona}.

**Episode Details:**
{dialog_text}
Selected Product: {selected_product_id}
Feedback: {feedback}

**Your task:** Provide the context from this episode that you would want a future agent to know. Focus on:

- What worked or didn't work in your approach.
- Key insights about user preferences or product selection.
- Any patterns you noticed that could help in similar situations.

**Instruction:** Write only the summary, no additional commentary.

Figure 13: LLM prompt variant for generating episode summaries.

---

**Prompts Enhancement for Chain-of-Thought (CoT)**

Let me think through this systematically:

- Customer preferences: [analyze what I know]
- Available products: [analyze the options]
- Best match: [reason about the best choice]
- Decision: [decide whether to ask or recommend]

**Let's reason step by step:**

1. What do I know about the customer so far?
2. What information am I still missing?
3. Based on this reasoning, what should I do next?

Before making your decision, think again: What are you unsure about regarding this customer? What questions should you ask next? Consider what additional information would help you make a better recommendation.

Think through each step carefully before responding.

Figure 14: Chain-of-Thought (CoT) enhancement appended to agent prompts to test the reasoning process.

## A.4 PLANNING AGENTS

In addition to the baseline recommender prompts described above, we introduce prompts designed to explicitly encourage planning behavior. These variants aim to test whether models can adopt more deliberate strategies for information gathering rather than defaulting to shallow heuristics.

The **Greedy** prompt (Figure 15) directs the agent to internally enumerate plausible candidate products and then select the single most informative clarifying question that would best differentiate among them.

The **POMDP** prompt (Figure 16) frames the interaction as a planning problem under uncertainty. Here, the agent is instructed to maintain a belief state over possible user preferences, evaluate the expected value of different candidate questions, and select the one with the highest information gain—even if its benefits only materialize in later turns.

Results are shown in Figure 8.

---

**Prompt for Greedy Questioning**

You are a product recommendation agent. Your goal is to find the best product for this user.

**Context:**
{context}
{feedback_context}

**INTERNAL REASONING (do not share with customer):**

- First, list all the possible products that you think the customer might like based on what you know so far.
- Then, think about what is the best question you could ask the customer to eliminate the most number of products from the list.

Your question should:

- Help you distinguish between the products you think the customer might like
- Focus on the most important decision factor that's still unclear
- Ask about preferences, needs, and requirements — NOT about specific products or product numbers

**CRITICAL OUTPUT FORMAT (MUST FOLLOW EXACTLY):**
QUESTION: [your question here]

---

Figure 15: Greedy prompt used to bias the recommender agent toward asking the most informative single question at each turn.

---

**Prompts for Dynamic Programming (POMDP) Questioning**

**System Message:** You are a product recommendation agent. Your goal is to find the best product for this user.

**Context:**
{context}
{feedback_context}

Think like a planner solving a **POMDP** with a **single terminal reward** from the score that the customer would assign to what you recommend. Note that you have {questions_remaining} turn(s) left.

1. **Maintain a belief state**—a probability distribution over possible customer preferences given past answers.
2. For each possible next question:
   - Predict how each possible answer will **update your belief**.
   - Estimate how that updated belief will affect your **final recommendation quality**.
   - Compute the **expected value of information (EVI)** for that question.
3. Choose the question with the **highest expected value**, even if it has no immediate payoff.

**Customer Interaction:** Ask the question that maximizes the expected value of information for your final recommendation.

**CRITICAL OUTPUT FORMAT (MUST FOLLOW EXACTLY):**
QUESTION: [your question here]

---

Figure 16: POMDP-inspired prompt that instructs the agent to choose the question with the highest Expected Value of Information.

## A.5 MANUAL QUESTIONING TESTS

To verify that poor agent performance is not due to an unsolvable environment, we introduce manual questioning experiments where a human designs the persona interactions. Figure 17 shows an exem-

plar run in interactive mode, where it can be noted that carefully chosen questions enable recovery of the best product with zero regret.

---

**An Example Manual Questions Experiment (Interactive Mode)**

**Running Manual Questions Experiment (Interactive Mode):**
Persona: 25905
Category: Office Racks & Displays
**=== Manual Questions Experiment (Interactive) ===**
Persona: 25905
Category: Office Racks & Displays
Model: gpt-4o, Feedback: persona

Category Office Racks & Displays: Max score 82.5, proceeding

> **Q:** How do you usually organize your desk at home or in the office?
> **A:** I organize my desk by keeping it tidy and functional, with my laptop in the center, a notepad and pen to the right, and any necessary documents or files neatly stacked on the left.

> **Q:** Do you prefer everything stored neatly in trays, or do you like quick vertical access to files and folders?
> **A:** I prefer everything stored neatly in trays.

> **Q:** How many papers, notebooks, or files do you typically keep on your desk at one time?
> **A:** Usually around three or four.

> **Q:** Do your kids or family members often use your desk space, or is it mainly your own?
> **A:** It's mainly my own.

> **Q:** Do you like darker neutral tones (black, brown, gray), or brighter finishes like gold or rose?
> **A:** I like darker neutral tones.

> **Q:** Do you want a compact organizer to save space, or a larger one with multiple tiers for capacity?
> **A:** A compact organizer to save space.

> **Q:** /done

**Products in Office Racks & Displays:** 26
Top 3 products by score:

>  1. SimpleHouseware Mesh Desk Organizer with Sliding D... (Score: 82.5)
>  2. Figella Leather Office File Rack Document Filing T... (Score: 80.0)
>  3. Febou 2 Pack Mail Organizer Desktop File Holder, 3... (Score: 80.0)

**Final recommendation:** Product 107921
Score: 82.5, Best: 82.5
Top1: True, Top3: True
Feedback: *Perfect! This is exactly what I was looking for. Great recommendation!*

Experiment completed successfully!
Chosen product rank: 1
Final regret: 0.0

---

Figure 17: An exemplar run of the manual questions experiment in interactive mode.

# B ABLATION STUDIES

We discuss additional ablation studies performed using the dataset. Overall, SOTA models do not exhibit the ability to learn across turns or episodes.

## B.1 VARYING USER FEEDBACK

Fixing the model to GPT-4o and holding all else constant, we evaluated the three feedback types described in Section 4.3 across 40 random seeds and 10 episodes. As shown in Figure 18, no

statistically significant differences were observed across feedback types, indicating that the form of feedback does not materially influence model performance.

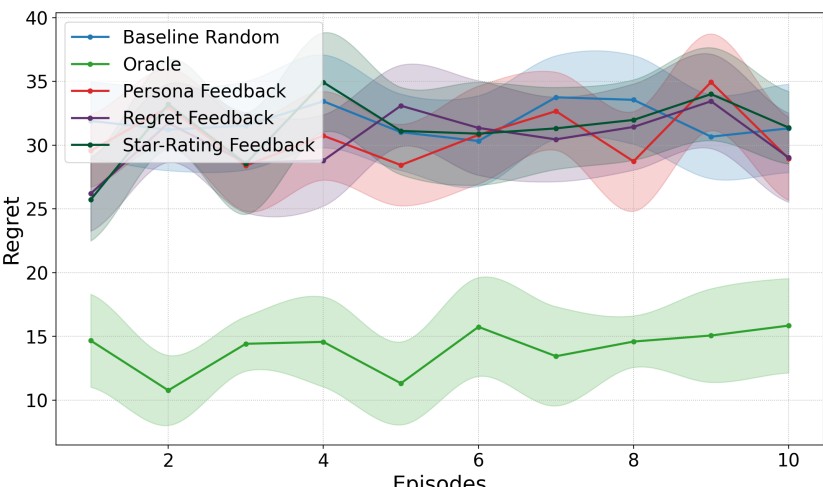

Figure 18: Feedback type did not improve model performance by any significant extent

## B.2 PROVIDING REASONING PROMPTS

Fixing the model to GPT-4o and holding all else constant, we also experimented with prompting strategies, such as inserting reasoning prompts (e.g., Think Again) before the agent issued a recommendation or question. As shown in Figure 19, these prompting tricks did not lead to statistically significant improvements, and performance remained indistinguishable from the no-prompting baseline.

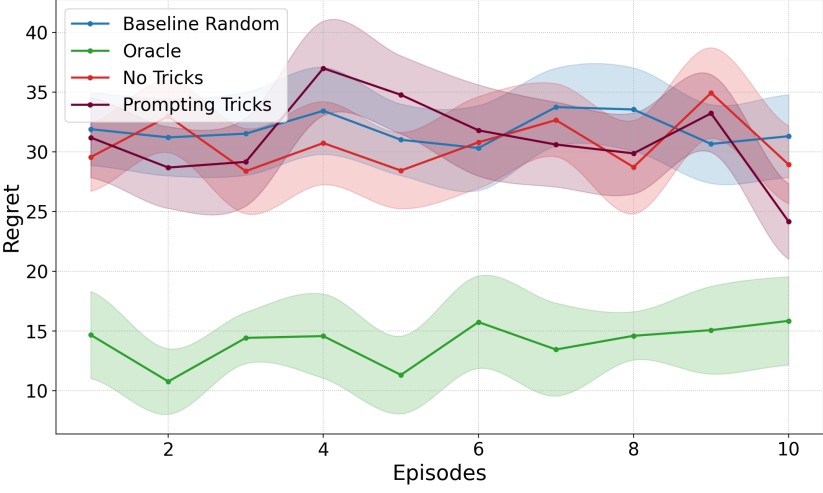

Figure 19: The presence and absence of prompting tricks did not impact model performance by any significant extent

## B.3 HOW TO UTILIZE PAST EXPERIENCES

Fixing the model to GPT-4o and holding all else constant, we further examined whether providing agents with access to their own context summaries could improve performance. In this variant, the recommender agent generated a summary of each episode, which was then carried forward into

subsequent interactions. As shown in Figure 20, this approach produced no statistically significant improvement over the baseline.

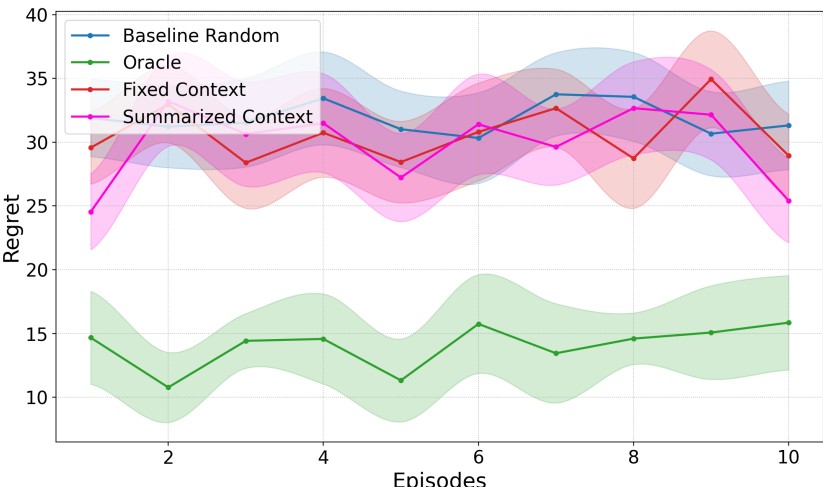

Figure 20: Context type did not improve model performance by any significant extent

## B.4 LEARNING ACROSS DIFFERENT USERS

Using GPT-4o with all other factors held constant, we tested whether varying product categories, personas, or both across episodes influenced model performance. As shown in Figure 23, none of these variants produced statistically significant differences.

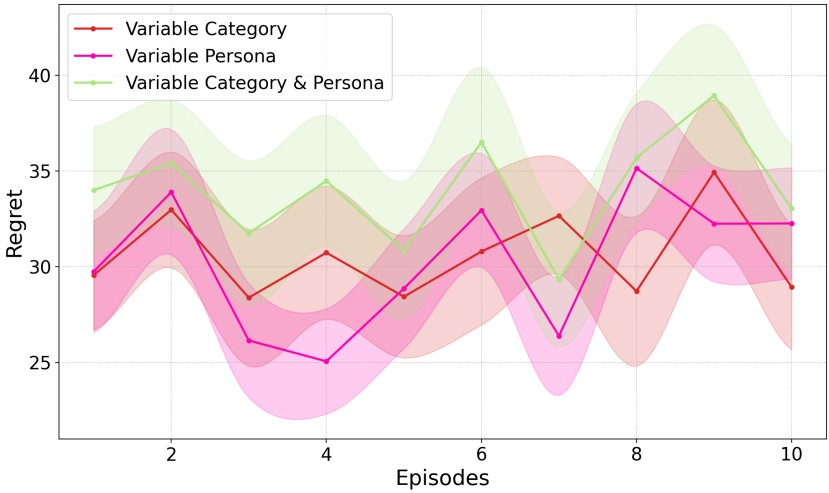

Figure 21: Whether there was variable personas, categories, or both did not impact model performance by any significant extent

## B.5 ECE

We prompted the agent, at every recommendation turn, to report confidences for five binary targets: (1) chosen item is top-1 for the persona, (2) top-5, and (3–5) regret $\leq$ 5, 10, 20, 30. For each target, we evaluated calibration per episode slice (Episodes 1, 5, and 10), pooling all turns across categories and seeds for that slice. We produced bin-wise accuracy-vs-confidence plots and summarized

misalignment with a single score. We ran the protocol under two prompting conditions, Baseline (no strategy) and DP-style planning, and display the full grid of episode × target results for each condition.

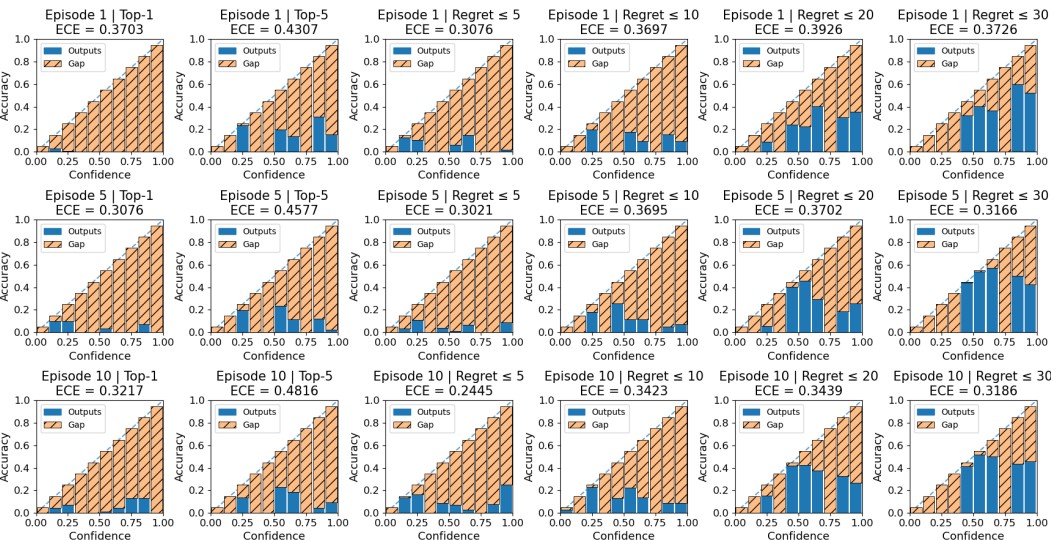

Figure 22: All ECE Runs using GPT-4o with No Strategy

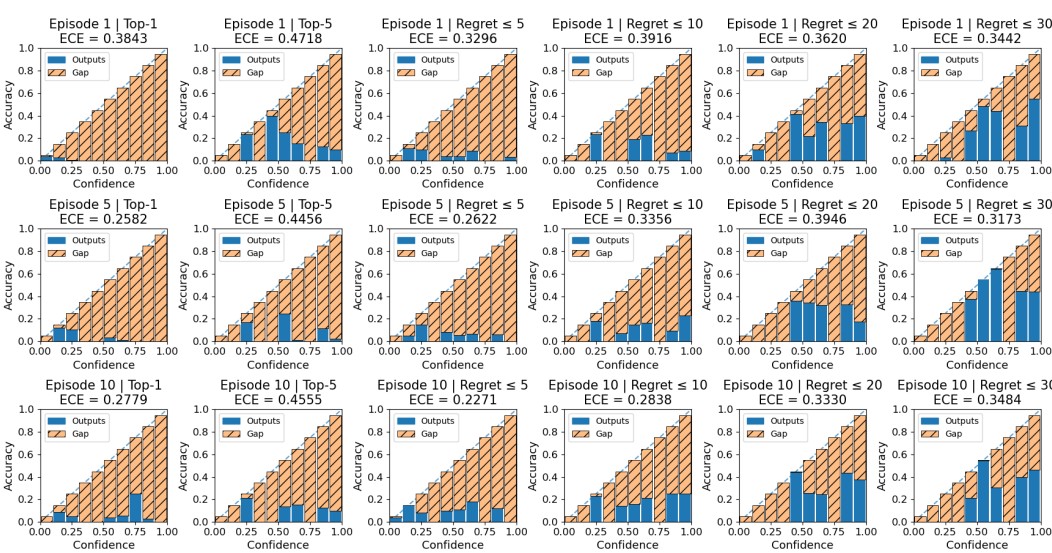

Figure 23: All ECE Runs using GPT-4o with DP-style Planning

## C  DATASET COLLECTION

**Collecting Products.** The Amazon Reviews dataset (Hou et al., 2024), consisting of 34 product categories, is set up such that one has to pick one of the 34 categories to sample products. Therefore, to preserve the original distribution of categories, we performed weighted random sampling over the categories. We provided the necessary scripts to draw more products in our codebase (Section 7).
**Matching Product Categories.** The first is inconsistent naming conventions; for example, a product's category path might be ...→Women → Shoes, Sandals, whereas our standardized path is ... → Women's Fashion → Women's Shoes → Women's Sandals. The second challenge

is structural discrepancy, where a product's path omits an intermediate level, such as `CDs&Vinyl` → `Blues`, while the canonical path is `CDs&Vinyl` → `Music Styles` → `Blues`. To solve these issues, our algorithm employs a two-pronged approach at each step of the path traversal:

- Fuzzy Matching for Naming Inconsistencies: To resolve differing names, we apply a series of prioritized fuzzy matching heuristics. For instance, the substring detection heuristic is precisely what allows the algorithm to map a product's simpler category `Sandals` to the more descriptive canonical term `Women's Sandals`. Other heuristics, like word subset validation (mapping Action Figures to Action & Toy Figures) and normalization (handling case and suffix differences), ensure robustness against a wide range of naming variations.

- Subtree Traversal for Structural Gaps: To handle missing intermediate levels, our algorithm does not merely search the immediate children of the last matched node. Instead, it performs a breadth-first search through the entire subtree of descendants. This is how it bridges structural gaps. In the example `CDs & Vinyl` → `Blues`, once the algorithm successfully matches the `CDs & Vinyl` node, it then searches all descendants for a `Blues` node. It will find the correct Blues node even though it is a grandchild (nested under `Music Styles`), effectively "skipping over" the missing level in the product's path data.

This matching enables us to successfully map about 70% of the sampled products to a valid path within the target hierarchy. Despite the flexibility of these heuristics, the overall assignment criteria remain strict: a product is considered successfully matched only when its entire category sequence corresponds to a valid, continuous path from the root.

**Final Database.** From the remaining category levels, we extracted individual category levels and flattened the hierarchical structure into a relationship database schema. The processed data was organized into three tables: (1) a products table containing product metadata, (2) a categories table, and (3) a product-category table implementing a many-to-many relationship between products and categories. From this database, we then pruned categories that were still too broad (e.g., Cooking & Baking) and would not be searched for by the typical user and niche categories that had fewer than 15 products. The final database consists of 71088 products, 2030 categories, and 100485 product-category links.

## D KNOWN ISSUES WITH LLMS

**Simulator Faithfulness** The customer simulator's intermediate responses are at times unfaithful to the persona's true underlying interests. Across the experiments, two main patterns of unfaithfulness were observed. First, in cases of consistent unfaithfulness, some personas consistently feign interest in the presented product category during the dialogue, only to reveal their true, unrelated intent in the final feedback. For instance, a persona might claim they are buying for a friend but, in the feedback stage, reveal personal annoyances with the product. Second, inconsistent faithfulness was observed in other personas (e.g., the woodworking enthusiast, No. 2601), who sometimes truthfully state their interests but at other times actively deceive the agent by fabricating plausible but false needs, such as inventing a specific camera model they do not own. In all cases, the simulators are designed to provide misleading or incomplete information, forcing the agent to learn from indirect signals and negative feedback.

**Impact of Reasoning Prompts and Behavior** It is not clear if the models' behavior materially changes with a reasoning prompt. Models, both with and without prompting tricks, sometimes ask a significantly higher volume of questions. This strategy can show signs of rigidity and inefficiency, as the model tends to ask many repetitive questions and relies heavily on a simple yes/no format. For instance, in the `Fabric Dyes` episode, the agent asks the exact same question three times consecutively. Similarly, models from the Gemini series exhibit their own specific rigid patterns. They frequently ask questions like,"Are you looking for a free option?" and then often proceed immediately to a recommendation. This behavior suggests the model is following a pre-determined conversational script rather than dynamically adapting to the user's needs, showing no interest in asking more questions and prematurely ending the conversation.

The reasoning process, as observed in the `thinking_block`, consists of a short textual monologue. It typically summarizes the user's explicitly stated needs from the dialogue and then outlines

a simple strategy for its next action, which is often a recommendation. The reasoning itself is sensible and logical based on the information the agent has at that moment. The agent correctly processes the user's statements and devises a rational plan. The failures observed are not due to flawed reasoning but are a direct result of the agent reasoning from the intentionally false or misleading premises provided by the unfaithful simulator.

## E    LLM USE

LLMs were used to aid writing, as well as for finding some of the related works.

