# OpenReview forum: "Benchmarking In-context Experiential Learning Through Repeated Product Recommendations"
_ICLR.cc/2026/Conference — Submitted to ICLR 2026_

### Official Review · Reviewer_Ebit · 2025-10-29

**Soundness:** 2
**Presentation:** 2
**Contribution:** 2
**Rating:** 2
**Confidence:** 4

**Summary:**

The paper proposes a benchmark for evaluating "In-Context Experiential Learning" (BIEL) through repeated product recommendations. The authors argue that current evaluations fail to capture agents' ability to adapt and improve based on previous interactions, especially in ambiguous environments. They introduce a dataset combining Amazon products, diverse user personas, and an LLM-based user simulator to assess how well agents can learn through experience and handle uncertainty.

**Strengths:**

1. The exploration of agent behavior in ambiguous scenarios is an important research question. The motivation to build a benchmark that measures the ability of agents to adapt to evolving environments is highly relevant.

2. Using recommendation systems as a testbed for in-context experiential learning is an intuitive choice. The complexity of user preferences and dynamic product landscapes provides a challenging environment for agent evaluation.

**Weaknesses:**

1. A core claim of the paper is that current evaluations fail to address ambiguity in agent tasks. However, the design of this benchmark does not clearly incorporate ambiguity in the environment. Instead, the focus appears to be on diversity, which does not fully capture the ambiguity in the agents' decision-making process.

2. The authenticity of the benchmark's behavior is questionable. LLMs are known to produce hallucinations, and since personas are generated by LLMs and used as user simulators, the reliability of the feedback cannot be guaranteed without additional verification or quality checks.

3. User interaction history is critical for user modeling in recommendation systems. The lack of this in the benchmark simplifies the recommendation scenario too much, potentially overlooking the complexity of real-world interactions that affect user behavior and preferences.

4. The paper tests a narrow set of models ( there are more model families can be explored such as Qwen, Kimi, DeepSeek, Doubao, GLM) and does not include reasoning models, which are more powerful. Additionally, the scale of the data tested seems insufficient for generalizable conclusions.

5. The experimental results do not provide sufficient analysis of the models' performance over time or detailed insights into the failures observed. A more comprehensive analysis of the models’ behavior and further experiments are needed to strengthen the claims.

**Questions:**

see weakness

---

> ### Author Response · Authors · 2025-11-21
>
> We thank the reviewer for highlighting the relevance of the model’s capability we wish to evaluate with BIEL, and are glad that you appreciate the use of recommendation setting to instantiate this evaluation.
>
> >**... ambiguity in agent tasks …**
>
> The main ambiguity in the task stems from the fact that the persona of the customer (the latent) is not revealed to the agent. Therefore, the agent needs to inquire specific questions to gather part of the latent relevant to the recommendation tasks in current and future episodes. As shown by the results, this ambiguity is already great enough that the models do not significantly outperform the simple baselines.
>
> >**... authenticity of the benchmark’s behaviour …**
>
> We hope the above general response adequately addresses the reviewer’s concern on authenticity of the benchmark, on both the realism and consistency fronts. Regarding the concern on LLM hallucinations, we have performed additional experiments involving 1300+ response generations to verify (manually and with LLM-as-judge) that the LLM simulator does not hallucinate given the persona description. (**$\geq \boldsymbol{93}$%** consistency. See [here](https://github.com/17my15/extra_experiments/blob/main/Extra%20Experiments/Persona_Response/Persona_Description_Consistency.md) and [here](https://github.com/17my15/extra_experiments/blob/main/Extra%20Experiments/Response_Consistency/Response_Consistency.md))
>
> >**... User interaction history is critical …**
>
> We completely agree with the reviewer that user interaction history is critical. Therefore, our benchmark focuses on measuring whether agents can actively gather and utilize such user interaction history. Note that prior customer interaction history is included in the context of the next recommendation task, so this complexity of the recommendation scenario is captured in our benchmark.
>
> >**... a narrow set of models … does not include reasoning models …**
>
> We have additionally included experiments using Qwen, Kimi, and Deepseek models, and found that they do not significantly outperform the baselines either (see [here](https://github.com/17my15/extra_experiments/blob/main/Extra%20Experiments/Supplementary%20Models/New_Models_Experiments_Analysis.md)). We would also like to highlight that we did benchmark state-of-the-art reasoning models such as Gemini-2.5-pro, Claude-Opus-4, and Claude-Sonnet-4 in the experiments.

---

### Official Review · Reviewer_PiVa · 2025-10-31

**Soundness:** 2
**Presentation:** 3
**Contribution:** 2
**Rating:** 4
**Confidence:** 3

**Summary:**

This paper proposes BIEL (Benchmark for In-context Experiential Learning), a benchmark designed to evaluate the ability of large language models to learn from experiential experiences in conversational product recommendations. Experiments on various proprietary and open-weight models reveal that current LLMs exhibit little to no improvement across rounds, highlighting the challenge of building agents capable of in-context experiential learning. While this paper is meaningful and well-written, it still lacks several necessary discussions.

**Strengths:**

**Timely discussion about the experiential learning problem in the recommendation scenarios.**
The paper highlights a crucial yet overlooked problem for current LLMs: in-context experimential learning. Moreover, this paper discusses experiential learning under the recommendation scenarios, which is practical and reasonable.

**Comprehensive experimental setup.**
BIEL provides a large-scale, systematically generated environment, spanning multiple user personas and product domains, allowing broad coverage for reproducibility and scalability.

**Good paper writing.**

The motivation of this paper is clear, and the whole paper is easy to understand.

**Weaknesses:**

**Lack of discussion on existing conversational recommender systems.**

A similar and standard research era in recommender systems is the conversational recommender system. However, this paper lacks the necessary discussion about this research era. The multi-turn recommendation paradigm is actually a common setting in conversational recommender systems, requiring discussion [1].

**Lack of sufficient analysis about the poor performance of existing advanced LLMs.**

Existing works demonstrate that advanced LLMs can perform well on multi-turn recommendation. However, this paper provides the opposite point of view about this. It would be necessary to discuss the failure of advanced LLMs, especially by comparing with existing works [2].

**Lack of discussion about the faithfulness of the user simulator.**

One crucial part of making the benchmark faithful enough comes from the design of the user simulator. Although this paper initializes these user simulators with real user profiles, it also requires a discussion about how close these simulated users actually align with real user behaviors and how to avoid information leakage when using LLMs as the simulator [3].

[1] Towards Deep Conversational Recommendations. NeurIPS 2018.

[2] Large language models as zero-shot conversational recommenders. CIKM 2023.

[3] How Reliable is Your Simulator? Analysis on the Limitations of Current LLM-based User Simulators for Conversational Recommendation. WWW 2024.

**Questions:**

Refer to the weakness part.

---

> ### Author Response · Authors · 2025-11-21
>
> We thank the reviewer for appreciating the value of the core capability that our benchmark aims to evaluate and for highlighting various strengths of the experimental setup.
>
> >**discussion on existing conversational recommender systems…**
>
> We thank the reviewer for bringing our attention to relevant literature. In the revised manuscript, we will use the extra page afforded to us to provide a thorough discussion of these works.
>
> >**... analysis about poor performance of existing advanced LLMs …**
>
> The work by He et al. (Large language models as zero-shot conversational recommenders. CIKM 2023) provides all previous conversation history to assess the model’s recommendation quality. In contrast, we assess whether the model can actively gather such relevant conversation history, and further *learn across episodes” of interactions. As depicted in Figure 1, the setting of He et al. would correspond to the *Full-Information* setting on the top left, significantly different from our setting on the right.
>
> We thank the reviewer for highlighting this relevant work, and will discuss our differences from this work in the revision.
>
> >**... faithfulness of user simulator**
>
> We agree with the reviewer’s point on realism of the simulator, and hope the general response above clarifies why our focus is on *consistency* of the simulator instead. Regarding information leakage, our simulator does not have access to the ground-truth product scoring when generating responses, so there is no risk of information leakage.

---

### Official Review · Reviewer_Vp1X · 2025-10-31

**Soundness:** 3
**Presentation:** 4
**Contribution:** 4
**Rating:** 8
**Confidence:** 4

**Summary:**

This paper introduces BIEL, a benchmark designed to evaluate large language models’ (LLMs) in-context experiential learning capabilities through repeated product recommendation tasks. The benchmark constructs realistic shopping episodes using Amazon product data, user personas, and an LLM-based interactive user simulator. Agents conduct multi-turn dialogues to infer latent user preferences and recommend products. The benchmark supports diverse settings, varying user and product dynamics, and measures performance via regret, question-asking behavior, and calibration metrics. Experiments on state-of-the-art models (GPT-4o, Gemini-2.5, Claude) show limited improvement across episodes and poor uncertainty calibration.

**Strengths:**

Strengths:
- This paper presents a novel task design for LLM study. The focus on learning across episodes is a great plus, as this is very important for practical deployment. However, this is largely unexplored.
- The dataset size is also large, including ~71K products, ~2K choice sets, and ~1M personas, which can enable scalable evaluation across diverse domains.
- This paper also provides a multi-model evaluation showing no meaningful improvement over episodes and poor uncertainty calibration. This points out an interesting direction for future research.
- The framework supports variable customer and category configurations. This can allow multiple experimental regimes.
- The work also provides more promises: human-constructed questions show that improvement is possible, and further validate the task difficulty.

**Weaknesses:**

Weakness:
- My biggest concern is that I could not find a user study or evaluation with real human preferences to validate the benchmark realism. I think this is critical to ground the benchmark in real human studies.
- It'd be great if the authors could discuss how the choice of regret, stars, and text feedback can impact the results.
- It'd be helpful to have pure human baselines. I understand that this can be costly, but at least mention this in the future work can be helpful to inform future studies.

**Questions:**

Please see the weakness points. The main concern is about the real human study for grounding.

---

> ### Author Response · Authors · 2025-11-21
>
> We thank the reviewer for the highly encouraging comments and assessment of our work, and are particularly glad that you appreciate the value of our experiment involving human-constructed questions.
>
> >**benchmark realism … pure human baselines…**
>
> We share the reviewer’s view that validating against real human preferences and collecting human baselines at scale, though outside of what our resources afford, would further enhance the work. We hope the general response above conveys why our focus has been on consistency of the benchmark. In the revised manuscript, we will thoroughly discuss these points as suggested by the reviewer.
>
> >**how the choice of regret, stars, and text feedback can impact the results**
>
> Empirical results show that none of the three types of feedback significantly impact the results, which suggests that the models are unable to utilize per-episode feedback in future recommendation. We will include a discussion on this interesting failure mode in the revised manuscript.

---

### Official Review · Reviewer_dbCB · 2025-10-31

**Soundness:** 2
**Presentation:** 3
**Contribution:** 2
**Rating:** 4
**Confidence:** 5

**Summary:**

The paper presents a benchmark targeted at learning to make product recommendations through multiple interactions with a single user or multiple users. The benchmark specifically targets learning over a series of interactions to move past the single step learning paradigm commonly used in RLHF for instruction following. The benchmark is designed to be interactive by relying on persona-conditioned LLM proxy humans that are able to answer questions the recommender agent poses or to react to the recommender agent's product recommendation. Performance measures rely primarily on persona-specific LL-Judge scores assigned to Amazon products, which is they used to compute regret. Several experiments with naive implementations of LLM agents that recommend products are used to demonstrate there is value to researching in this space as personalized product recommendation over the course of customer interactions does not improve with the number of interactions.

**Strengths:**

- The paper addresses an important problem, which is the ability to conduct personalized LLM research in interactive settings.
- The benchmark is grounded in real Amazon product data.
- There is a large number of different "people" and products to evaluate a LLM on.
- The environment is interactive and dynamic, so the proxy humans are impacted by the decisions of the learning agent.
- Based on the presented experiments, the problem needs active work as LLMs don't inherently have the ability to improve recommendations through increased interactions with the user.
- Despite a few typos, the paper is easy to read and follow.

**Weaknesses:**

- The main weakness of this paper is that the personas and the LLM-as-a-Judge scoring approach are not well validated.
     - The paper the personas come from was used to identify limitations with current approaches to creating personas by identifying issues and biases with the set of 1M personas introduced, which are used in this paper. These personas were found to be biased, which means the they are not overly unique, limiting the impact of having 1M personas.
     - The only measure of the LLM-as-a-Judge's scoring performance is whether the judge is consistent. However, consistent and accurate/realistic are two different things. Therefore, it is not clear how reliable the benchmark's performance metrics are. For example, the performance of LLM judges applied to the PRISM dataset (https://arxiv.org/pdf/2404.16019) varies greatly depending on the participant predicting for and ranges from an accuracy of 0 - 100.
     - Without well validated personas and scoring metrics, it is difficult to understand the expected size of the sim2real gap, which means it is challenging to know how likely results developed on the benchmark will generalize to learnings that hold up in the real world.
- The benchmark set up ignores the important aspect that human users may not be willing to answer 20 questions to help an agent make a product recommendations, and the willingness of a customer to answer questions will be customer specific. This reduces the realism of the human proxies, and therefore misses a key aspect of applying learnings from this benchmark to the real world.
- The benchmark does not appear to have any mechanism by which to introduce noise in the learning signal. As the real world and humans especially are full of noise, this makes it tricky to understand how well learnings from the benchmark will generalize to the real world.
- Small things:
     - Figure 6 does not appear to be referenced in the main body of the paper
     - ORACLE results should be plotted in Figure 8 to make the performance gap very clear.

**Questions:**

- The second use case for the benchmark is not making complete sense to me: "..the agent repeatedly sells a fixed set of products to a stream of new customers. Here, the goal shifts to identifying how these products compare relative to another across the diverse distribution of customers." It is not clear to me how this differs from the first, personalization, use case. Can you please elaborate?
- It is not completely clear the role humans played in the "Robustness check" detailed in Append A.5. Did the human play the role of agent, including making a product recommendation? What was used as the ground truth for the true product to recommend?
- In Section 5.1 "Comparing Base Models", what is the learning set up? Are the LLM agents simply conditioned on the interaction history with the proxy human customers?

---

> ### Author Response · Authors · 2025-11-21
>
> We thank the reviewer for appreciating the importance and the richness of the setting. We are particularly grateful for your thoughtful discussion on realism versus consistency, and fully agree with your concerns on the former. We hope the general response above adequately clarifies why our focus is on consistency instead. We have incorporated this discussion into the new manuscript, and sincerely appreciate you for bringing up this crucial point.
>
> >**... may not be willing to answer 20 questions …**
>
> We agree with the reviewer that users likely will not answer 20 questions. Echoing the general response, this is a design choice that mainly aims to give models plenty of opportunities to gather experience. We note that under the framework proposed, one can easily limit the evaluation to a few questions to reflect a more realistic setting.
>
> >**Figure 6… ORACLE should be plotted in Figure 8 …**
>
> We thank the reviewer for catching these issues. They have been corrected in the revised version.
>
> >**second use case for the benchmark…**
>
> In the first (personalization) setting, the choice set varies across episodes while the customer remains fixed. In the second setting, the choice set is fixed throughout, and the customers vary across episodes.
> For example, consider a typical salesperson in a clothing retail store. Over a given period (e.g., a season), the set of products offered remains largely unchanged, while the salesperson encounters a stream of heterogeneous customers. In this scenario, the recommender must develop hypotheses about the key dimensions along which customers form preferences within the fixed choice set (e.g. color, style, etc.), which inform the agent’s best questioning strategy. Note that at the high level, this also measures the agent's ability to actively explore latent shared across episodes.
>
> We have included this clarification to the revised manuscript.
>
> >**... role humans played…**
>
> We apologize for the confusion. Human only *asked* the questions, but the final recommendation, ground truth for the best product to recommend, and the responses to the questions, are all made by the LLMs in accordance with the benchmark’s setting. Hence, this experiment demonstrates that the task is solvable with expert’s questions, with all else held fixed.
> >**... Comparing Base Models… what is the learning set up…**
>
> Yes, the recommender agent is conditioned on the whole interaction history, including that from previous episodes.

---

> > ### Comment · Reviewer_dbCB · 2025-11-25
> > **Acknowledgement of Author Rebuttal**
> >
> > I have read the authors’ responses, and appreciate their feedback and clarification. However, I disagree with the authors about the importance of some degree of realism in a benchmark that centers on what is a heavily studied, human-centered, real world task for which algorithm are developed for real world applications on benchmarks like what is introduced in the paper.
> >
> > For the authors I see one of two possible paths forward:
> >
> > - use personas not already identified as problematic in the literature and validate the accuracy and diversity of the judges or
> >
> > - switch the task away from one so heavily entangled with a real world problem to focus on the authors’ target problem of learning from experiential information. Some examples include 20 Questions, Werewolf, and Wordle.
> >
> > After reading the other reviews, and to reflect the strength of my concern about the release of a benchmark without any expectation of realism in the human proxies nor evaluation measures when based on the recommender task, I am lowering my score.

---

### Author Response · Authors · 2025-11-21
**General Response**

We sincerely thank the reviewers for their thoughtful feedback and careful evaluation of our work. Particularly, we are grateful that the reviewers appreciate the importance of the proposed setting (dbCB, Vp1X, PiVa, Ebit), the richness of the experimental setup (dbCB, Vp1X, PiVa), and the writing quality (dbCB, Vp1X, PiVa). A main concern shared by all reviewers is about the *realism* of the benchmark. In light of this, we would like to take the opportunity to emphasize the main goal and contribution of this work, and particularly, clarify why we regard *consistency*, as opposed to *realism*, to be the key desideratum for the benchmark.


As all reviewers appreciate, the work’s primary objective is to evaluate the **experiential learning** capability of LLM agents. Rather than viewing LLMs as mere reservoirs of knowledge, able to solve any task in a zero-shot manner (impossible for tasks with uncertainty), we instead view in-context learning LLMs as algorithms: entities that can actively gather relevant experiences and consistently improve *across episodes* by leveraging prior experiences. As Reviewers Vp1X, PiVa, and Ebit note, this viewpoint highlights a natural yet curiously underexplored aspect of intelligence. Hence, designing a benchmark that isolates and measures this capability is a meaningful contribution to the broader effort of evaluating and developing intelligent agents.

Agnostic to specific instantiation of the benchmark, the key requisite for such a benchmark is the presence of a **consistent** latent shared across all episodes. This latent must be reflected in both (i) the experiences to which agents have access, and (ii) the evaluation of agents’ performance. Only under such consistency can agents reasonably be expected to benefit from prior experiences and exhibit episode-over-episode improvement. Our main contribution is to instantiate this design principle in a recommendation setting and to extensively validate that the consistency property holds:
- Consistency between persona specification and product scoring, as shown by high performance of the oracle baseline
- Consistency between persona-conditioned response and scoring, as shown by:
  - our hand-crafted experiment in Appendix A.5 (as pointed out by Reviewer Vp1X)
  - **[NEW]** our analysis on a trajectory found by GPT-4o that utilized customer responses in prior episodes to gradually improve in recommendation quality (see [here](https://github.com/17my15/extra_experiments/blob/main/Extra%20Experiments/Ideal%20Trajectory/Analysis.md))
  - **[NEW]** our new experiments where the recommender agents perform better than simple baselines with improved simulator’s prompts (see [here](https://github.com/17my15/extra_experiments/blob/main/Extra%20Experiments/Updated%20Experimentation/Analysis.md))
- **[NEW]** Consistency between persona specification and responses, as shown by an extensive examination of over 300 responses (see [here](https://github.com/17my15/extra_experiments/blob/main/Extra%20Experiments/Persona_Response/Persona_Description_Consistency.md))
- **[NEW]** Consistency between different sampled responses from the simulator (see [here](https://github.com/17my15/extra_experiments/blob/main/Extra%20Experiments/Response_Consistency/Response_Consistency.md))

Regarding *realism*, we fully agree with the reviewers’ concerns. Indeed, it is unclear to what extent performance on our benchmark transfers to real-world recommendation scenarios, given the complexity and heterogeneity of human behavior that persona-conditioned LLMs may not fully capture. Crucially, however, realism is not critical to our goal of evaluating the in-context experiential learning ability of LLM agents. As the only evaluation framework focusing on this capability, we believe our benchmark already offers substantial value even without achieving full realism.

That said, we agree with the reviewers that realism would further strengthen the impact of the benchmark. To the extent our resources afford, we deliberately design the benchmark with realism in mind, using rich product data, carefully curated choice sets and a diverse personas pool. Further establishing realism at scale, however, requires access to a large volume of user purchasing data accompanied by user information, typically available only to major corporations and pose privacy concerns to open-sourced academic studies. We also note that there are substantial and ongoing efforts aimed at improving the fidelity with which LLMs simulate human behavior, and we are optimistic that advances on this front will further enhance the value of our evaluation framework. We sincerely thank the reviewers for raising this fine point, and will revise the manuscript to include a more extensive discussion of the benchmark's limitations.

---

### Meta-Review · Area_Chair_DbDQ · 2025-12-29

**Summary:**

This paper studies an interesting problem. All of the reviewers brought up concerns about realism, since both the personas and scoring metrics are based on LLMs. Even the reviewer with an accept score said that real human studies are "critical." I acknowledge the author response about isolating and measuring experiential learning specifically. However, I think including some validation of transferability to real settings is important for a benchmarking paper.

**Reviewer Concerns:**

I believe several concerns unrelated to realism have been addressed, and that the realism concerns have not been addressed.

**Reviewer Scores:**

Reviewer dbCB said that they were lowering their score, and I wouldn't be surprised if reviewer Vp1X also chose to lower their score.

---

### Decision · Program_Chairs · 2026-01-26

Reject